# Healing Properties of Epidermal Growth Factor and Tocotrienol-Rich Fraction in Deep Partial-Thickness Experimental Burn Wounds

**DOI:** 10.3390/antiox9020130

**Published:** 2020-02-03

**Authors:** Hui-fang Guo, Roslida Abd Hamid, Razana Mohd Ali, Sui Kiat Chang, Mohammed Habibur Rahman, Zaida Zainal, Huzwah Khaza’ai

**Affiliations:** 1Department of Biology, Faculty of Basic Science, Chengde Medical University, Chengde 067000, China; ghf1428@163.com; 2Department of Biomedical Sciences, Faculty of Medicine and Health Sciences, Universiti Putra Malaysia, Serdang 43400, Selangor, Malaysia; roslida@upm.edu.my; 3Department of Pathology, Faculty of Medicine and Health Sciences, Universiti Putra Malaysia, Serdang 43400, Selangor, Malaysia; razanamo@upm.edu.my; 4Key Laboratory of Plant Resources Conservation and Sustainable Utilization, Guangdong Provincial Key Laboratory of Applied Botany, South China Botanical Garden, Chinese Academy of Sciences, Guangzhou 510650, China; changsk@scbg.ac.cn; 5Department of Pathology, Faculty of Veterinary Science, Bangladesh Agricultural University, Mymengsingh 2202, Bangladesh; rahmanmdhabib@gmail.com; 6Nutrition Unit, Product Development and Advisory Services Division, Malaysian Palm Oil Board, Bandar Baru Bangi 43000, Selangor, Malaysia; zaida@mpob.gov.my

**Keywords:** burn wound, epidermal growth factor (EGF), tocotrienol-rich fraction (TRF), partial-thickness burn, wound healing

## Abstract

*Background:* An experimental study was undertaken to determine the efficacy of the epidermal growth factor (EGF) with tocotrienol-rich fraction (TRF) cream in the wound-healing process on skin with deep partial-thickness burn in rats. *Methods:* A total of 180 Sprague-Dawley rats were randomly divided into six groups of six each and were: untreated control, treated with Silverdin^®^ cream, base cream, base cream with c% EGF, base cream with 3% TRF or base cream with c% EGF and 3% TRF, respectively. Creams were applied once daily for 21 consecutive days. Six animals from each group were sacrificed using anaesthetic overdose on the third, seventh, 11th, 14th and 21st day post-burn. Skin tissues with the wound to be examined were excised for macroscopic and microscopic evaluation and biochemical analyses. *Results:* EGF + TRF formulation decreased the number of neutrophils, lymphocytes and myofibroblasts post-burn. However, no effects on the number of adipose cells in the healing process were recorded. In addition, lipid peroxidation and nitrite production were found to be reduced post-burn, reducing oxidative stress. Conclusions: Results of the present study indicate that the addition of EGF with TRF have ameliorating effects on deep-partial thickness burn healing parameters.

## 1. Introduction

Burns are defined as skin lesions and are caused mainly by heat or other acute trauma, including scalds, contact with heating elements and flames. Burn injuries lead to different degrees of damages [1] to the blood vessels. According to the depth of the injured tissue, burns are classified as superficial, partial-thickness and full-thickness burns [2]. Burns that affect only the epidermis are known as superficial or first-degree burns [3]. When lesions penetrate into the dermis, they are denoted as partial-thickness or second-degree burns and further classified as superficial or deep partial-thickness burns [4]. Deep partial-thickness burns can cause deep dermis damage where the skin is drier, the sensation of the skin tends to be weakened and the hair is easy to be sloughed off. Deep partial-thickness burns heal slowly and are accompanied by scar formation and potential functional loss [4]. 

The use of topical applications of growth factor therapy have assisted in minimizing the incidence of burn wound scars and long-term healing [5]. The epidermal growth factor (EGF) is a small peptide with 53 amino acids [6]. The role of the EGF in dermal wound healing has been extensively studied. It stimulates the proliferation of epithelial cells, endothelial cells and fibroblasts and accelerates burn wound healing processes [7,8]. In recent years, vitamin E family members tocotrienols and tocopherols have been widely known for their health benefits. Vitamin E extracted from crude palm oil is mainly composed of a mixture of 70% tocotrienols and 30% tocopherols and referred to as a tocotrienol-rich fraction (TRF) [9,10]. The many distinctive roles of tocotrienols in cancer, inflammation, neuroprotection and metabolic syndromes exemplify that tocotrienols have significant implications for clinical use [10,11,12,13,14]. Tocotrienols identified as antioxidants have been well-known in the field of dermatology for many years [15]. Zampieri et al. [16] confirmed the positive effect of topical vitamin E in surgical wound repair. Although the effect of TRF has been shown in cellular biological experiments, to the best of our knowledge, studies using TRF in burn wound healing is very limited. The development of drugs/plant-derived product preparations containing numerous phytochemicals possessing various bioactive properties aimed at promoting wound healing is necessary. 

Therefore, the present study was undertaken to record macroscopic changes (gross appearance, clinical evaluation and wound contraction rate) and microscopic changes (histological scoring and cell counting) of skin tissues with deep partial-thickness burns after being treated with EGF + TRF formulation over time. It is hypothesized that the EGF + TRF formulation has synergistic effects in treating deep partial-thickness burns. Besides, the levels of malondialdehyde (MDA) and nitrite in the wound tissues of all the groups over time post-burn were measured to evaluate the effects of the EGF + TRF formulation on the oxidative stress level in rats with deep partial-thickness burns.

## 2. Materials and Methods 

### 2.1. Materials

SSD (Silverdin^®^ cream) was purchased from Sunward Pharmaceutical (Tampoi, Johor, Malaysia) and TRF (Gold Tri E70) was purchased from Sime Darby, Kuala Lumpur, Malaysia. Pure EGF drug (GF144) was purchased from Merck Milipore (Darmstadt, Germany). Xylazine and ketamine were obtained from Troy Laboratories (Glendenning, Australia); Tramadol was purchased from CCM Duo Pharma (Kuala Lumpur, Malaysia); and Depilatory cream (VEET™) was purchased from Reckitt Benckiser (Sydney, NSW, Australia). Paraffin, Eosin and DPX mounting medium were purchased from Merck (Darmstadt, Germany). Hematoxylin was purchased from Thermo Shandon (Bartlesville, OK, USA) and 10% Formaldehyde, ethanol and xylene were purchased from Systerm (Kuala Lumpur, Malaysia). TBARS Assay Kit (Cat. no.10009055) was purchased from Cayman Chemical (Ann Arbor, MI, USA). Griess reagent and sodium nitrite were purchased from Sigma Chemical (St. Louis, MO, USA).

### 2.2. Animals

Male Sprague-Dawley rats (*n* = 180) weighing 250 ± 50 g were used in this study. The rats from each group were housed individually in Perspex cages with glass rods in the base to facilitate collection of food spillage and faeces and fed ad libitum on standard laboratory rat chow. Water was given ad libitum. All the rats were housed in a temperature-controlled (25 ± 1 °C) environment with a 12-h light/dark cycle. The rats were randomly divided into six groups: untreated (UN control group), treated with Silverdin^®^ cream (SSD group), base cream (BC group), base cream with c% EGF (EGF group) (the concentration of EGF was undisclosed for intellectual property rights), base cream with 3% TRF (TRF group) or base cream with c% EGF and 3% TRF (EGF + TRF group), respectively. For this experiment, groups of six rats were used. All animal experiments were approved by the Institutional Animal Care and Use Committee, Universiti Putra Malaysia, Malaysia (AUP Number: R095/2014).

The following experimental groups were used and shown in Table 1.

In order to assess the effect of EGF + TRF formulation on deep partial-thickness burns in the healing process, six animals from each group were killed by diethyl ether overdose on days three, seven, 11, 14 and 21 post-burn, and the wound tissues were excised immediately and were divided into two parts. The first part was fixed in 10% buffered neural formalin for microscopic assessment (histopathological analysis and cell counting). The second part was frozen at −80 °C for oxidative stress analyses. 

### 2.3. Preparation of Base Cream, EGF Cream, TRF Cream and EGF + TRF Cream

Base cream containing sorbitol (C_6_H_14_O_6_) butylene glycol (C_4_H_10_O_2_) alcohol (Ethanol), glycol stearate (C_20_H_40_O_3_) PEG 100 stearate (C_34_H_70_O_9_) stearyl alcohol (CH_3_(CH_2_)_16_CH_2_OH) polysorbate 20 (C_58_H_114_O_26_), dimetricone (C_2_H_6_OSi), carbon (charcoal), aminomethyl propanol (C_4_H_11_NO) phenoxyethanol (C_8_H_10_O_2_), methylparaben (C_8_H_8_O_3_) and propylparaben (C_10_H_12_O_3_) was made by Dr. Huzwah Khaza’ai from the Department of Biomedical Sciences, Universiti Putra, Malaysia. The exact ratio of the ingredients in the base cream was not revealed due to the issue of intellectual property rights. EGF cream was made by the mixture of c grams EGF per 100 g base cream. c% EGF was the best concentration after optimisation in our laboratory. To protect intellectual property rights, the incorporation of EGF was shown as the lowercase “c”. TRF cream was prepared by the mixture of 3 g TRF per 100 g base cream. The concentration of 3% TRF was used according to a previous study [17]. EGF + TRF cream was made by the mixture of c g EGF and 3 g TRF per 100 g base cream. All the above-mentioned creams were stirred using a sterile disposable spatula in a fume cupboard and stored in a commercial cream container at 4 °C before further analysis.

### 2.4. Burns Procedure

The procedure for burns was carried out according to the method established by Guo et al. [18] and was essentially as follows: First, the rats were acclimatised to laboratory conditions for one week prior to commencement of the experiment. Then, the animals were weighed and anaesthetised by intramuscular injection of ketamine (75 mg/kg) and xylazine (15 mg/kg). The dorsum of the already anaesthetised animals was shaved by electric clippers and depilated with a commercial depilatory cream (VEETTM; Reckitt Benckiser, Sydney, NSW, Australia) to remove the remaining stubble that may interfere in creating a desired and uniform burn wound. Burn injuries to the dorsum were created using a heating device that was designed in the laboratory and fitted with a 20-mm-wide circular aluminium head device. The wounds were created in 10 s. The pressure exerted on the skin was 300 g and each wound was created at five-minute intervals to allow the aluminium head to yield the required temperature. Once the wound creation was finished, treated rats received tramadol, a narcotic-like pain reliever (12.5 mg/kg body weight) subcutaneously twice daily to alleviate pain from burns. 

### 2.5. Treatment of Wounds

After the burn induction, the wound was washed by spraying the sterile saline solution and dried using sterile gauze. The cream was applied evenly over the wound once daily for a consecutive 21 days. In addition to the control group, all the groups were treated following the same procedures. During the entire experimental period, the wounds were kept uncovered.

### 2.6. Gross Evaluation

Images of the burnt area of each animal were captured by digital camera (Canon SX240 HS, Canon Inc., Tokyo, Japan) on days 1, 3, 7, 11, 14 and 21 post-burn. A ruler was placed on the side of wound to act as a known scale between pixels, which were useful for the calculation of wound area by Image J1.49v software (National Institutes of Health, Bethesda, MD, USA). Gross changes of the wound, including the general appearance, size, time for crust formation and detachment, were recorded. 

### 2.7. Clinical Evaluation

Burn healing was observed for 21 days post-burn. Clinical evaluation parameters were recorded, including duration of oedema, days required for sloughing off of the crusts and subsequent re-epithelialization. Oedema was caused by excess fluid in the body tissues. The duration of oedema was the time from the presence to the absence of the oedema. The sloughing off of the crust was referred to as the time from the day of induction of burn lesions to the day that the crust was completely sloughed off from the wound. Sloughing off of the crusts, leaving no raw wound area, was considered to be the end point of complete re-epithelialisation. The time from the induction of burn to the day that the epidermis was completely healed was taken as the period of re-epithelialisation [19].

### 2.8. Wound Contraction Rate

After digitalization, the wound area was analysed by Image J 1.49v software (National Institutes of Health, Bethesda, MD, USA). The wound contraction rate was expressed as the percentage change in the original wound area using the following formula:Wound contraction rate = (original wound area − wound area on that particular day)/Original wound area × 100%

The original wound area was the area of the aluminium head and the specific day wound area was the area of wound measured on that particular day.

### 2.9. Microscopic Analyses

After being sacrificed, blocks of half of the damaged skin tissues from the rats were excised and fixed in 10% buffered neutral formalin. The samples were then processed through graded alcohol, chloroform, infiltrated with molten wax and embedded in paraffin in the Paraffin Embedding Station (Leica EG1600, Leica, Germany). Sections were cut at 5 µm thickness with Rotary Microtome (Leica RM 2135, Leica, Germany) and dried at room temperature at least for one day. Prior to dyeing, the slides were placed in an oven (Venticell, Newtown, Germany) at 60°C for an hour to remove the paraffin. After that, sections were stained with Hematoxylin & Eosin (H&E), Abbey Color, PA, USA. Histological changes in stained sections were studied under an optical microscope (Leica Microsystems AG, Germany). To investigate the histological changes in the wound-healing process, histological scoring was carried out on the H&E staining slides of skin tissue samples taken on days 3, 7, 11, 14 and 21 post-burn in all the groups according to the histological scoring grades for the epidermis and dermis. The histological scoring is attached as Appendix A.

### 2.10. Cell Count 

The digital micrographs were captured approximately halfway between the superficial aspect of the wound and the dermal junction, with a magnification of 40×. Six photographs from each sample were used for counting. Different types of cells, including adipose cells, neutrophils, lymphocytes and myofibroblasts, were counted. The average number of cells counted from six micrographs was used for statistical analysis.

### 2.11. Determination of Oxidative Stress

#### 2.11.1. Tissue Preparation

Prior to the analysis, the samples were taken out from the −80 °C freezer and placed on ice. After that, the crust was removed from skin tissues where the rest of the tissues were weighed and washed thoroughly with physiological saline (4 °C). The tissue samples were then placed in PBS solution with a ratio of 1:10 and then homogenized on ice. After centrifugation at 4000× *g* for 30 min, the supernatant was used for the measurements of MDA and nitrite.

#### 2.11.2. TBARS Assay

Tissue samples or standard MDA solutions (100 μL each) and 100 μL SDS solution were added to a 5 mL vial. They were shaken to mix with 4 mL of TBA solution. The vials were then placed in boiling water for an hour. After that, the vials were removed immediately and put in an ice bath to stop the reaction. The vials were then centrifuged at 1600× *g* for 10 min at 4 °C and incubated at room temperature for 30 min. Subsequently, the supernatant from each vial (150 μL) was loaded into a transparent 96-well colorimetric plate. The absorbance value was read with an enzyme-linked immunosorbent reader (Biotek EL×800, Winooski, VT, USA) at a wavelength of 520 nm. MDA values for each sample were calculated using the standard calibration curve. 

#### 2.11.3. Determination of Nitrite

The present study utilized the Griess assay, which quantified the amount of nitrite and the enzymatic reduction product of nitrate. Briefly, 50 μL of Griess reagent was added to all wells in a transparent 96-well colorimetric plate and incubated for a period of 10 min at room temperature. The absorbance values of the pink reaction product were read at 530 nm using an enzyme-linked immunosorbent assay (ELISA) plate reader (Biotek EL × 800, Winooski, VT, USA), and double-distilled H_2_O was used as a blank.

### 2.12. Statistical Analyses

All data were expressed as mean ± S.E.M. Data for wound contraction rates in all the groups were analysed by repeated measures of two-way ANOVA with Bonferroni’s multiple comparisons test. The rest of the data were analysed by two-way ANOVA with Bonferroni’s multiple comparisons test. All statistical analysis was completed with Statistical Package for the Social Sciences (SPSS) version 22 software (SPSS Inc., Chicago, IL, USA). A *p*-value of less than 0.05 was considered to be statistically significant.

## 3. Results

### 3.1. Macroscopic Evaluation

The images of deep partial-thickness burn wounds on days 0, 1, 3, 7, 11, 14, and 21 post-burn are shown in Figure 1. The post-burn wounds appeared to be round, which was due to the burner’s aluminium head. Oedema resulted in uniform white wounds. No blister was noticed, and the edges between the wound areas and the normal skin were well-demarcated. On day one post-burn, oedema remained, and the smallest crust formation was noticed in the area around the wound. Until the third day, the entire area of the burned skin was covered with a solid layer of crust. On the seventh day post-burn, the crust became dark and became smaller in size. By the 11th day, the crust became drier and smaller where the edges of the crust were found to be separated from the wound. On the 14th day, the crust was completely sloughed off the skin, forming a second discreet crust. Upon reaching day 21, the wound area was found to be smaller; however, it failed to achieve complete epithelialisation.

Immediately after burns, the wounds were round in shape and turned white in colour with oedema. From day one to day three, oedema started receding and the wounds were covered with a firm crust. From day seven to day 14, the crust on the wounds became drier and smaller and finally sloughed off from the wounds, and then second discreet crusts were formed. By day 21 post-burn, the discreet crusts were found to be present but much smaller in size. The wound areas were small, but complete epithelialisation was not shown.

### 3.2. Clinical Evaluation

Table 2 listed the clinical evaluation features, including duration of oedema, sloughing off of crusts in days and the period of re-epithelialization. Immediately after burn, the skin was pearly white with oedema. After oedema was completed, the wound was covered with a layer of brown and solid crust. The oedema persisted for three days for all the groups. Then, the crust became dry and finally sloughed off the wound. The sloughing off of crust for the UN group was 14.40 ± 0.65 days, slightly longer than the rest of the groups, which was around 13 days. After the crust sloughed off from the wound, the second discrete crust was formed and then sloughed off the wound. By that time, epithelialisation was completed. The time to complete epithelialisation in the UN group was more than 21 days, higher than that found in the BC, EGF, TRF, EGF + TRF or SSD groups. 

### 3.3. Wound Contraction Rate

The effect of EGF + TRF formulation on burn wound areas was shown in Figure 2. The rate of wound contraction in all experimental groups increased in a time-dependent manner. On the third day post-burn, the wound contraction rate was around 20% in each group, with the smallest in the UN group and the highest in the EGF group. There was no significant difference (*p* > 0.05) between groups. On the seventh day post-burn, the wound contraction in the EGF + TRF group was 51.51 ± 4.43%, which appeared to be significantly higher (*p* < 0.05) than the corresponding UN (26.94 ± 3.08%) and the SSD group (37.11 ± 2.48%). In addition, the TRF group also showed faster wound contraction rates compared to those of the UN group (*p* < 0.05). The wound contraction rates of the EGF + TRF group (64.26 ± 3.33%) and TRF group (58.19 ± 6.28%) were again significantly higher (*p* < 0.05) than the UN group (42.79 ± 3.12%) on the 11th day post-burn. On 14th day post-burn, the wound contraction rates were found to be increased to more than 70% in all the groups. However, there were no significant differences (*p* > 0.05) between groups. On day 21 post-burn, the wound contraction reached 100% in the EGF + TRF, EGF, TRF, BC and SSD groups. From the above, the wound contraction rates were significantly higher (*p* < 0.05) on days seven and 11 post-burn in the EGF + TRF group, and similar phenomenon was observed in the TRF group.

### 3.4. Histological Analysis

The epidermal histological scores of all experimental groups over time post-burn were summarized in Table 3. On the third day post-burn, all the groups scored zero, and this was due to deep partial-thickness burns leading to complete loss of the epidermis. On the seventh day post-burn, the score of the EGF + TRF group was slightly higher than other groups. Nonetheless, on the 11th day post-burn, all the groups scored between 2.50 and 3.00. On the 14th day post-burn, the histological score was found to be significantly higher (*p* < 0.05) in the EGF + TRF group (6.00 ± 0.52) than those of the UN (3.67 ± 0.42) and SSD groups (4.29 ± 0.42), while the rest of the groups had a histological score of about 4.00. This indicated that the epidermis of the EGF + TRF group was completely restored, while the rest remained in the stage of epidermal migration. On the 21st day post-burn, the score in the EGF + TRF group was 6.67 ± 0.33, which was significantly higher (*p* < 0.05) than those of the UN group (4.75 ± 0.25). Meanwhile, the SSD, EGF, TRF and BC groups had scores of around 6.00. By and large, the EGF + TRF formulation had accelerated epithelialisation on days 14 and 21 post-burn.

The dermal histological scores of each group over time post-burn were shown in Table 4. On the third day post-burn, the dermal score for all the groups was zero. On the seventh day post-burn, the score of the dermis was around 1.00 in the UN, BC, EGF and SSD groups. On the other hand, it was 2.29 ± 0.29 in the EGF + TRF group and 2.80 ± 0.20 in the TRF group, which was significantly higher (*p* < 0.05) than those of the UN and SSD groups, indicating smaller proportions of adipose cells and higher proportions of inflammatory cells and fibroblasts in the EGF + TRF and TRF groups. On the 11th day post-burn, the score was similar in all the groups (*p* > 0.05). On the 14th day post-burn, the dermal histological score in the EGF + TRF group was 6.00 ± 0.26, which was significantly higher (*p* < 0.05) than the corresponding UN (4.00 ± 0.37) and SSD groups (4.67 ± 0.21). By day 21 post-burn, there was again a significant difference (*p* < 0.05) between the EGF + TRF group (6.67 ± 0.33) and the UN group (5.25 ± 0.25), while the scores for the rest of the groups were around 6.00. Thus, the EGF + TRF formulation appeared to accelerate dermal healing on days 7, 14 and 21 post-burn significantly (*p* < 0.05).

### 3.5. The Number of Adipose Cells in All the Groups over Time Post-Burn

The number of adipose cells in each group against time post-burn was shown in Figure 3. On the third day post-burn, the number of adipose cells was significantly higher (*p* < 0.05) in all the groups, ranging from 9 to 12. While on the seventh day post-burn, the number of adipose cells was found to be significantly decreased (*p* < 0.05) in all the groups. It was highest in the UN and BC groups, which were 3.67 ± 1.08 and 3.39 ± 1.37, respectively. On the other hand, the number of adipose cells in the EGF + TRF and TRF groups were the lowest. There was statistically significant difference (*p* < 0.05) between the TRF and the UN groups. On the 11th day post-burn, a small amount of adipose cells were found in the UN, BC and EGF groups, yet no adipose cells were found in the EGF + TRF, TRF and SSD groups. In addition, adipose cells were not found in all the groups on the 14th and 21st days post-burn. It was revealed that the change in the number of adipose cells in each group against time post-burn was similar, since all of which were the maximum on the third day post-burn and then decreased with time. There was no significant difference (*p* > 0.05) in the EGF + TRF treatment group when compared to those of the untreated group.

### 3.6. The Number of Neutrophils in All the Groups over Time Post-Burn

The number of neutrophils in each group over time post-burn was shown in Figure 4. On the third day post-burn, the number of neutrophils ranged from three to nine in all the groups, then shifted on the seventh day and decreased slightly on the 11th day post-burn. By the 14th day post-burn, the number of neutrophils in the UN, BC, EGF and SSD groups was significantly increased (*p* < 0.05); however, it was found to be decreased to about five, which was significantly lower (*p* < 0.05) in the EGF + TRF group, as compared to those of the UN and SSD groups. On day 21 post-burn, the number of neutrophils in all the groups decreased significantly (*p* < 0.05) to a lower level. The number of neutrophils in the EGF + TRF treatment group maintained relatively low levels throughout the healing process, especially on the 14th day post-burn, suggesting that EGF + TRF treatment could reduce the inflammatory response at later stages of healing.

### 3.7. The Number of Lymphocytes in All the Groups over Time Post-Burn

The number of lymphocytes in all the groups over time post-burn was shown in Figure 5. On the third day post-burn, there was almost no lymphocytes observed in all the groups. By the seventh day post-burn, the number of lymphocytes were significantly increased (*p* < 0.05) in almost all the groups, ranging from three to six. On the 11th day post-burn, the number shifted in all the groups. By the 14th day post-burn, the number of lymphocytes was significantly increased (*p* < 0.05) in the BC group and slightly increased in the EGF, TRF and SSD groups. However, it was significantly (*p* < 0.05) decreased in the EGF + TRF group (3.00 ± 0.40), as compared to those of the SSD group. On the 21st day post-burn, the lymphocytes were found to be significantly depressed (*p* < 0.05) in all the groups.

### 3.8. The Number of Myofibroblasts in All the Groups over Time Post-Burn

The number of myofibroblasts in each group against time post-burn was summarized in Figure 6. The results have shown a similar pattern in all the groups with an increase in cell numbers from day three to day 11, with a maximum number on day 11, and then showed a decreased from day 21 post-burn. On the third day post-burn, myofibroblasts were not detected in all the groups. However, on the seventh day post-burn, a significant increase (*p* < 0.05) in the number of myofibroblasts were observed in all the groups. The number of myofibroblasts in the EGF + TRF and TRF groups were found to be 44.38 ± 2.56 and 36.27 ± 1.63, respectively, and were significantly higher (*p* < 0.05) than those in the UN (18.78 ± 5.28) and SSD (18.78 ± 5.59) groups. On the 11th day post-burn, all the groups showed a maximum number of myofibroblasts. The maximum number was 51.50 ± 5.12 in the UN group, which was significantly higher (*p* < 0.05) than the corresponding EGF (33.50 ± 4.16) and SSD groups (33.00 ± 5.08). On the 14th day post-burn, the number of myofibroblasts decreased in all the groups. A significant shift (*p* < 0.05) in the number of myofibroblasts was found in the EGF + TRF (21.67 ± 2.61), EGF (30.00 ± 1.95) and SSD (31.22 ± 2.88) groups than in the corresponding UN group (46.33 ± 3.49). On the 21st day post-burn, the number of myofibroblasts continued to decrease and reached the lowest level among all the groups.

### 3.9. Measurement of Lipid Peroxidation

The level of MDA, a final product of lipid peroxidation, was measured in all the groups over days post-burn and shown in Figure 7. The MDA content on the third day post-burn was found to be the highest in all the groups, ranging from 5.89 to 6.77. On the seventh day post-burn, however, the MDA content was found to be significantly decreased (*p* < 0.05) in all the groups, except in the BC group. In addition, the MDA content in the TRF group (4.02 ± 0.07) was significantly lower (*p* < 0.05) than those of the UN group (5.60 ± 0.20). The level of MDA in all the groups continued to be depressed on the 11th day post-burn. On the 14th day post-burn, in contrast, the MDA level was slightly increased in the SSD, BC, EGF and TRF groups, while substantially increased in the UN group (*p* < 0.05). However, it was significantly decreased in the EGF + TRF group (*p* < 0.05). Compared to those of the UN group, the MDA levels in the EGF + TRF and TRF groups were found to be significantly lower (*p* < 0.05), especially in the EGF + TRF group. The smallest amount of the MDA content was found in all the groups on the 21st day post-burn. In the EGF + TRF and TRF groups, the content of the MDA was again significantly lower (*p* < 0.05) than those of the UN group. 

### 3.10. Measurement of Nitrite

The content of nitrite in all the groups over time post-burn was shown in Figure 8. On the third day post-burn, the content of nitrite ranged from 30.86–35.18, where there was no significant difference (*p* > 0.05) between groups. On the seventh day post-burn, the content of nitrite was significantly increased (*p* < 0.05) in the EGF + TRF (42.70 ± 0.80), EGF (40.83 ± 0.91) and TRF groups (41.66 ± 0.96) and slightly increased in the rest of the groups. On the 11th and 14th days post-burn, the level of nitrite varied in all the groups. However, a significant decrease (*p* < 0.05) of nitrite content was found in the EGF + TRF group (34.92 ± 1.35), as compared to the UN group (40.79 ± 1.19) on day 14 post-burn. By the 21st day post-burn, the nitrite level significantly decreased (*p* < 0.05) to the lowest in all the groups. A significant lower (*p* < 0.05) content of nitrite was found in the EGF + TRF (21.64 ± 0.47) and TRF (23.35 ± 0.70) groups, as compared to the UN group (29.61 ± 0.57).

## 4. Discussion

In the present study, the effect of the EGF + TRF formulation on deep partial-thickness burns was investigated on the 3rd, 7th, 11th, 14th and 21st days post-burn at gross and microscopic, as well as biochemical, assessments. On the third day post-burn, the wounds were found to be covered by layers of crusts in all the groups. At the histological level, although the crusts were on the dermis, the adhesions were quite loose. Therefore, they were usually lost during the staining process and could not be observed. In addition, the denudation of the epidermis was observed in all the groups, indicating consistent deep partial-thickness burns. In the dermis, neutrophils and adipose cells accounted for a large proportion in the process of wound healing. Neutrophils were active at the initial stage of wound healing [20]. Once in the wound sites, neutrophils debrided the wounds through the active oxygen intermediates, nitric oxide and activated lysozyme enzymes; in concert, neutrophils elaborated cytokines, lymphokines and growth factors to regulate wound healing [21]. Thus, at this stage, the majority of the cells were of this kind.

On the seventh day post-burn, the wounds entered into proliferative phases. In all the groups, the crusts were tightly attached to the wounds. Under the microscope, a close connection between the crusts and the dermis was observed, with very distinct boundaries. Histological scores of the epidermis in the EGF + TRF group were slightly higher than the rest of the groups, indicating faster epithelialisation. It may be mentioned here that the histological scores of the dermis in the EGF + TRF group were significantly higher (*p* < 0.05) than the corresponding UN group, which was supported by the results obtained from cell counting. On day seven post-burn, fewer numbers of adipose cells, higher numbers of neutrophils and significantly higher (*p* < 0.05) numbers of myofibroblasts were observed in the EGF + TRF group than the corresponding UN group. These changes suggested that the EGF + TRF formulation may have accelerated dermal healing. As a matter of coincidence, the significant increase (*p* < 0.05) in wound contraction rates were also recorded in the EGF + TRF group on the same day. It is highly likely that it may have been due to the increased number of myofibroblasts in the dermis that was responsible for providing the force for the wound contractions.

Taken together, the effects of the EGF + TRF formulation in accelerating wound healing on the seventh day post-burn were confirmed at the gross and microscopic levels. Several workers [22,23] reported beneficial and stimulatory effects of the EGF on keratinocytes, fibroblasts and vascular endothelial cells in the past. Thus, the possibility of beneficial effects of the EGF to augment myofibroblastic proliferation in the EGF + TRF formulation have again proved to be true.

On the 11th day post-burn, the crusts were harder and smaller but still attached to the wounds. Sections staining with H&E revealed that there was mild to moderate epithelialisation of the wounds. Besides, the dermal histological scores were around three for all the groups, indicating few adipose cells, many inflammatory cells and few fibroblasts in the dermis, which was also supported by the results of cell counting. By day 11 post-burn, few adipose cells were present in the UN, BC and EGF groups. However, none of them were observed in the EGF + TRF, TRF and SSD groups. Furthermore, the number of neutrophils were inconsistent in number in all the groups but still maintained at high levels. What is more, the maximum number of myofibroblasts was found in all the groups, which participated in both collagen synthesis and wound contraction [24].

On the 14th day post-burn, the crusts were found to be detached from the wounds in all the groups. Specifically, the average crust-falling day in the EGF + TRF group was 12.6, a few days earlier than the UN group, indicating an accelerated wound healing in the EGF + TRF group, which was supported by histological observations. On day 14 post-burn, when the moderate-to-high epidermal migration was observed in the UN group, complete epithelialisation had already been achieved in the EGF + TRF group, suggesting that the EGF + TRF treatment had significantly accelerated the rate of epithelialisation. The reason for the speeding epidermal formation of the EGF + TRF group may have been due to the early separation of the crusts. 

From the above, when the crusts were separated from the wounds, the wounds were able to fully come in contact with the cream, allowing the moistening of the wounds by the cream. Concomitantly, the significantly higher (*p* < 0.05) dermal histological scores were also observed in the EGF + TRF group, indicating less inflammatory cells and more fibroblasts in the dermis, which was again supported at the cellular level. By day 14, the number of neutrophils in the EGF + TRF group was significantly lower (*p* < 0.05) than those recorded from the UN group, and the number of myofibroblasts was also significantly decreased (*p* < 0.05). It was reported that, with the progression of wound healing, myofibroblasts were differentiated to fibroblasts or disappeared by cell apoptosis [25]; thus, the decreased myofibroblasts were possibly transformed into the fibroblasts and eventually led to the increase in the number of fibroblasts. 

On day 21, the epithelialisation was achieved in the EGF + TRF greater than the corresponding UN group. This again implied the effects of the EGF + TRF treatment in accelerating wound healing. Histological evaluations of the EGF + TRF-treated wounds also showed more rapid developments of the skin tissues. By day 21 post-burn, in the EGF + TRF group, the epidermis was found to be protruded into the dermis and formed rete ridge, which was absent in the rest of the groups. In the dermis, the formation of immature hair follicles was only observed in the EGF + TRF group. Thus, the EGF + TRF formulation accelerated wound healing, in comparison with other treatments. 

The number of myofibroblasts decreased to the lowest level in all the groups, which was important for re-modelling of the extracellular matrix, which might take a longer time than has been presumed. In addition, the number of neutrophils and lymphocytes were also lower. Unlike myofibroblasts, neutrophils played an important role at the beginning of wound healing. A variety of chemicals and enzymes were released by neutrophils to kill microbes. However, when the levels of these substances were unusually high in the extracellular matrix, the tissues might be damaged [21]. Moreover, neutrophils might delay wound closure by secreting substances that accelerated keratinocyte differentiation, rather than proliferation [20]. Therefore, after completion of the task, neutrophils usually disappeared due to cell apoptosis or were phagocytised by macrophages. Lymphocytes are generally involved in chronic inflammatory conditions [26]. At the end of wound healing, only small amounts of lymphocytes were observed in all the groups, indicating no chronic inflammation in the wounds had occurred.

Reactive oxygen species (ROS) are produced by all cells during normal metabolic processes and maintained at relatively low concentrations. However, in the pathological condition, the production of ROS is elevated. In wounds, an increased amount of ROS is produced by NADPH oxidase, an enzyme complex highly expressed in inflammatory cells [27]. Traditionally, ROS drives cell-mediated defence responses in wound healing. Neutrophils and macrophages utilise their reactive and destructive properties to attack invading pathogens and eventually kill them to protect the host. Studies have shown that low levels of ROS play a key role in the healing process of the wound [28]. It was reported that low concentrations of ROS promoted the proliferation and migration of keratinocytes during the proliferative phase [29]. Despite the positive effect of ROS, excess ROS in the wound may damage to the surrounding tissues through the oxidation of cellular macromolecules and cause damage to wound repairs [30].

In the present study, MDA levels were significantly decreased in the EGF + TRF treated wounds on days 14 and 21 post-burn. Interestingly, the reduced MDA levels were also observed in the TRF group on days 14 and 21 post-burn, which suggested that the EGF + TRF cream may reduce the level of MDA due to the antioxidant function of TRF. Vitamin E is responsible for the removal of peroxyl radical intermediates in lipid peroxidation and the protection of polyunsaturated fatty acids present in cell membranes [31]. Studies have also shown that tocotrienols from palm oil have the ability to inhibit protein oxidative damage in a time and concentration-dependent manner [32]. Thus, in the cream EGF + TRF, TRF successfully scavenged on peroxyl radicals at the later stage of wound healing.

In recent years, the role of NO in wound healing has been disputed [33]. Accumulating evidences have indicated the positive effects of NO on wound healing at metabolic and gene expression levels [34,35]. However, NO has also been shown to be closely related to the deleterious effects of peroxynitrite during the wound healing process by interaction with the superoxide radical. In addition, NO can upregulate the transcription of nuclear factor kappa B (NF-kB) to trigger inflammatory signalling pathways. The expression of NO is closely related to inflammation. Although there is no direct study, wounds with continuous inflammatory responses are likely to produce large amounts of NO [36]. In addition, the regulation of iNOS is mainly by inflammatory cytokines. Thus, as long as inflammation exists, NO will be maintained at high levels. In fact, high levels of NO have already been a marker for the treatment of inflammatory disorders [37]. 

Based on the findings in the present study, at the early stage of wound healing, inflammation existed due to the need to destroy bacteria, foreign particles, damaged tissue and the release of growth factors for cell migration and proliferation. Therefore, the level of NO in all the groups was high. In the midst of wound healing, inflammation still existed for the formation of granulation tissue. Therefore, the NO level was still high, although slightly reduced. However, at the final stage of wound healing, the number of inflammatory cells such as neutrophils were reduced in all the groups, indicating the end of the inflammatory condition. 

In parallel, the NO level was also restored to its physiological level in all the groups at the same time. The current study demonstrated that the NO level was significantly reduced on the 14th and 21st days post-burn in the EGF + TRF group, indicating that the EGF + TRF treatment could reduce the oxidative stress at the later stage of wound healing, therefore reducing inflammation in the wound area. The EGF promotes the proliferation of epithelial cells, endothelial cells and fibroblasts and accelerates the burn wound healing process [7,8]. Hence, it promotes tissue remodelling. Meanwhile, TRF from palm oil has a strong antioxidant activity that functions to alleviate oxidative stress by reducing lipid peroxidation and nitrite production post-burn in the burn wounds. This shows the synergistic mechanism of the combination of the EGF and TRF in the burn wounds healing processes of this study. These results are in agreement with a previously published study demonstrating the synergistic mechanism of vitamin E and EGF in protecting dermal fibroblasts in response to heat stress [38]. Collectively, the present findings provide clear evidence to suggest the potential benefits of the phyto-nutrients, like tocotrienol from oil palm TRF, to be useful as remedial agents in post-burn wound healing.

## 5. Conclusions

The gross/macroscopic evaluations revealed that the EGF + TRF formulation accelerated wound contractions post-burn and enhanced the epithelialisation in wound healing. Histological assessments showed the EGF + TRF formulation to treat burn wounds had clearly established the healing process of the epidermis post-burn and promoted the healing of the dermis post-burn. In addition, the results of cell counting confirmed that the EGF + TRF formulation decreased the number of neutrophils, lymphocytes and myofibroblasts post-burn, while showing no effects on the number of adipose cells in the healing process. It may be mentioned here that the EGF + TRF cream decreased oxidative stress by reducing lipid peroxidation and nitrite production post-burn. For the first time, our results have established the fact that TRF can be considered as an alternative approach to accelerate post-burn wound healing. Hence, further investigation is warranted to clarify the role(s) of TRF on post-burn wound healing in human subjects.

## Figures and Tables

**Figure 1 antioxidants-09-00130-f001:**
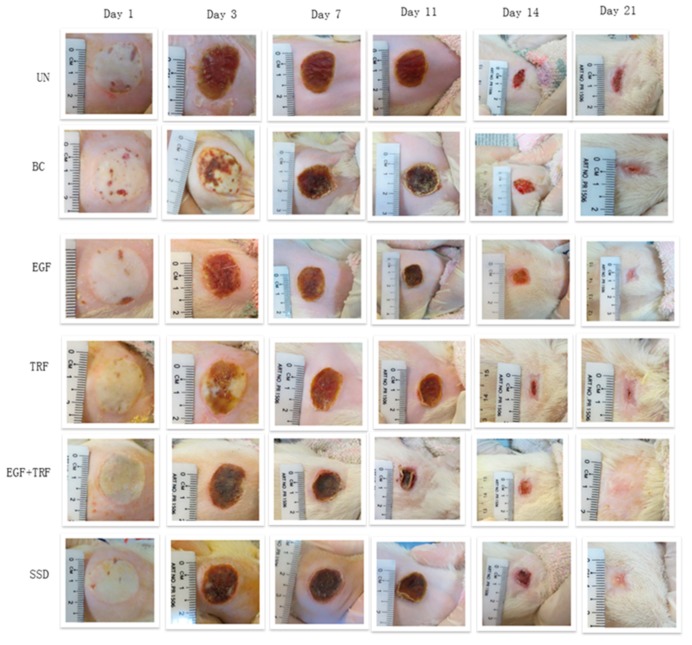
Macroscopic changes of deep partial-thickness burn wounds over time in Spraque-Dawley rats. UN: without any treatment, BC: treated with base cream, Epidermal growth factor (EGF): treated with base cream containing c% EGF, Tocotrienol-rich fraction (TRF): treated with base cream containing 3% TRF, EGF + TRF: treated with base cream containing both c% EGF and 3% TRF, and Silverdin^®^ cream (SSD): treated with SSD cream.

**Figure 2 antioxidants-09-00130-f002:**
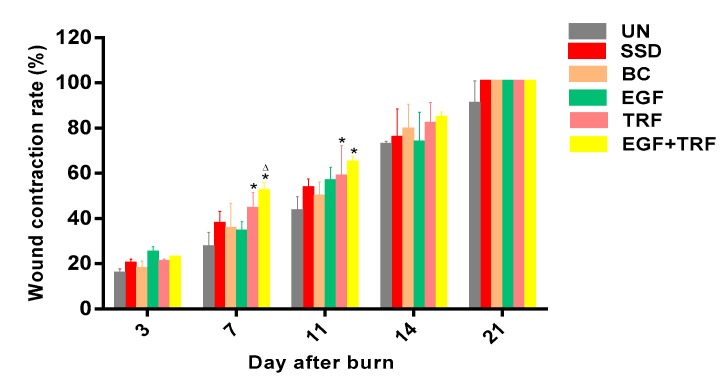
Wound contraction rates in all the groups over time post-burn. Data were expressed as mean ± S.E.M. * *p* < 0.05 compared with those of the UN group on the respective day and ^∆^
*p* < 0.05 compared with those of the SSD group on the respective day.

**Figure 3 antioxidants-09-00130-f003:**
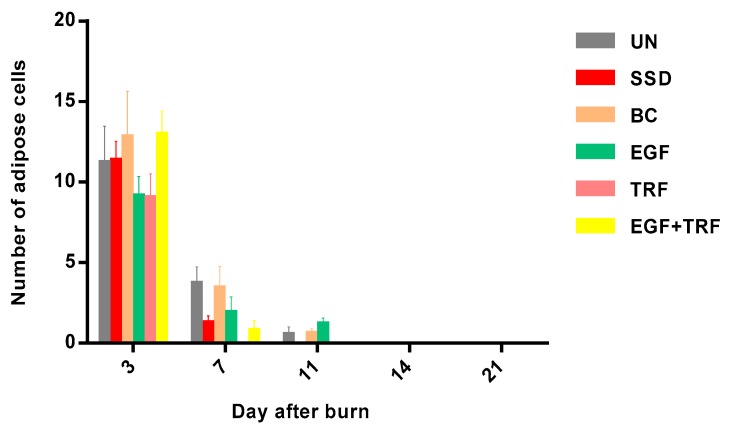
Number of adipose cells in all the groups over time post-burn. Note: Values calculated over a 21-day trial and expressed as mean ± S.E.M. for six rats.

**Figure 4 antioxidants-09-00130-f004:**
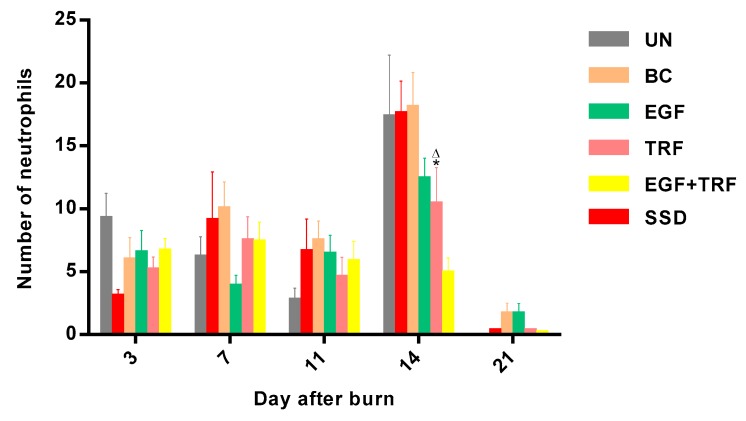
Number of neutrophils in all the groups over time post-burn. Note: Values calculated over a 21-day trial and expressed as mean ± S.E.M. for six rats. * *p* < 0.05 compared with those of the UN group on the respective day and ^∆^
*p* < 0.05 compared with those of the SSD group on the respective day.

**Figure 5 antioxidants-09-00130-f005:**
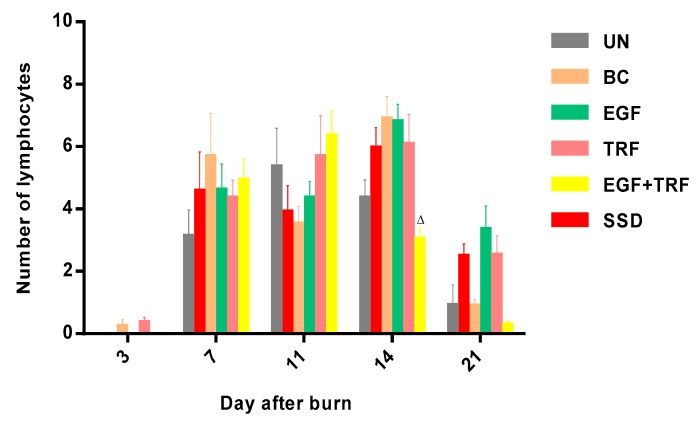
The number of lymphocytes in all the groups over time post-burn. Note: Values calculated over a 21-day trial and expressed as mean ± S.E.M. for six rats. ^∆^
*p* < 0.05 compared with those of the SSD group on the respective day.

**Figure 6 antioxidants-09-00130-f006:**
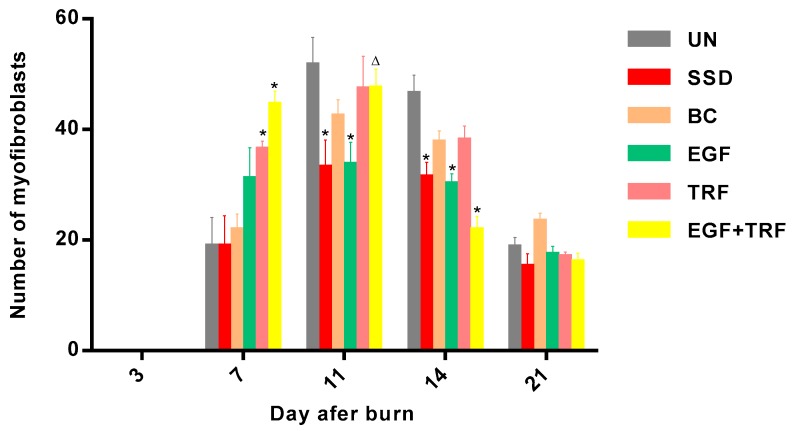
The number of myofibroblasts in all the groups over time post-burn. Note: Values calculated over a 21-day trial and expressed as mean ± S.E.M. for six rats. * *p* < 0.05 compared with those of the UN group on the respective day and ^∆^
*p* < 0.05 compared with those of the SSD group on the respective day.

**Figure 7 antioxidants-09-00130-f007:**
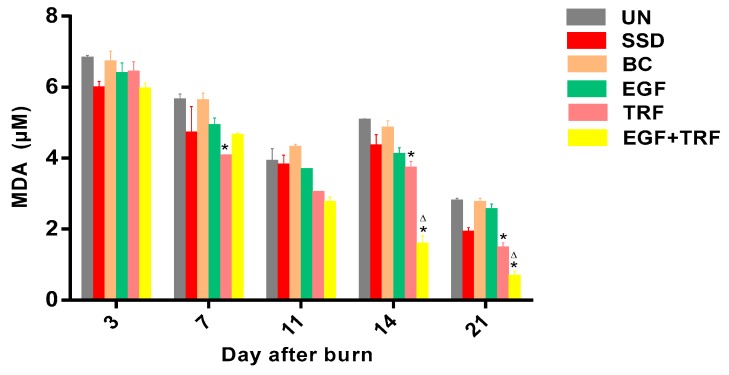
The malondialdehyde (MDA) content in all the groups over time post-burn. Data were expressed as mean ± S.E.M. * *p* < 0.05 compared with those of the UN group on the respective day and ^∆^
*p* < 0.05 compared with those of the SSD group on the respective day.

**Figure 8 antioxidants-09-00130-f008:**
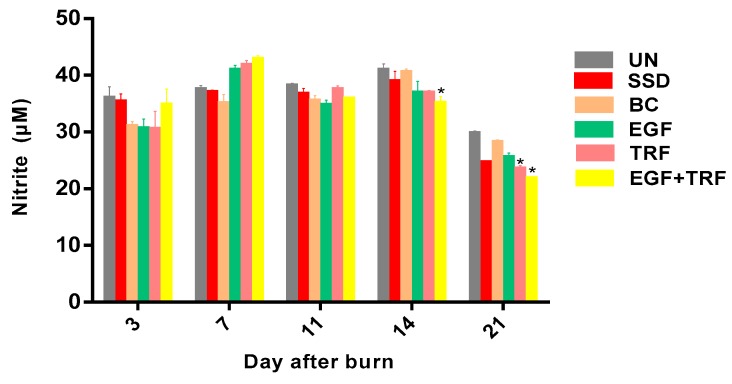
Nitrite content in all the groups over time post-burn. Data were expressed as mean ± S.D. * *p* < 0.05 compared with those of the UN group on the respective day.

**Table 1 antioxidants-09-00130-t001:** The experimental groups.

Group Codes	Treatment
UN	Burnt but were not treated
SSD	Burnt and were treated with Silverdin^®^ cream
BC	Burnt and were treated with base cream
Epidermal Growth Factor (EGF)	Burnt and were treated with base cream and c% EGF
Tocotrienol-rich Fraction (TRF)	Burnt and were treated with base cream and 3%TRF
EGF + TRF	Burnt and were treated with base cream, c% EGF and 3%TRF

**Table 2 antioxidants-09-00130-t002:** Clinical evaluations of wounds in all the groups during the healing process.

Groups	Period of Oedema (Day)	Fall of Crusts (Day)	Period of Epithelialisation (Day)
UN	3.00 ± 0.00	14.40 ± 0.65	>21.00
BC	3.00 ± 0.00	13.81 ± 0.56	20.00 ± 0.21
EGF	3.00 ± 0.00	13.50 ± 0.62	19.60 ± 0.21
TRF	3.00 ± 0.00	13.10 ± 0.53	19.40 ± 0.18
EGF + TRF	3.00 ± 0.00	12.60 ± 0.00	18.40 ± 0.18
SSD	3.00 ± 0.00	13.11 ± 0.61	19.40 ± 0.20

**Table 3 antioxidants-09-00130-t003:** Histological scores of the epidermis in all the groups over time post-burn.

Groups	Histological Scores of the Epidermis				
	Day 3	Day 7	Day 11	Day 14	Day 21
UN	0.00 ± 0.00	1.00 ± 0.00	2.50 ± 0.50 ^a^	3.67 ± 0.42 ^ab^	4.75 ± 0.25 ^abc^
BC	0.00 ± 0.00	1.17 ± 0.17 ^a^	2.80 ± 0.20 ^ab^	4.00 ± 0.45 ^ab^	5.50 ± 0.50 ^abcd^
EGF	0.00 ± 0.00	1.00 ± 0.00	3.00 ± 0.00 ^ab^	4.00 ± 0.37 ^ab^	5.80 ± 0.49 ^abcd^
TRF	0.00 ± 0.00	1.25 ± 0.25 ^a^	2.67 ± 0.33 ^a^	4.00 ± 0.37 ^ab^	6.00 ± 0.71 ^abcd^
EGF + TRF	0.00 ± 0.00	1.71 ± 0.18 ^a^	2.60 ± 0.25 ^a^	6.00 ± 0.52 *^∆abc^	6.67 ± 0.33 *^abc^
SSD	0.00 ± 0.00	1.00 ± 0.00	2.67 ± 0.33 ^ab^	4.29 ± 0.42 ^abc^	6.00 ± 0.55 ^abcd^

Note: Values calculated over a 21-day trial and expressed as mean ± S.E.M. for six rats. * *p* < 0.05 compared to the UN group on the respective day and ^∆^
*p* < 0.05 compared to the SSD group on the respective day. ^a^
*p* < 0.05, ^b^
*p* < 0.05, ^c^
*p* < 0.05 and ^d^
*p* < 0.05 compared to those recorded on days 3, 7, 11 and 14, respectively. Values expressed in a row with distinct superscript differ significantly, at least to a level of 95% confidence (*p* ≤ 0.05).

**Table 4 antioxidants-09-00130-t004:** Histological scores of the dermis in all the groups over time post-burn.

Groups	Histological Scores of the Dermis				
	Day 3	Day 7	Day 11	Day 14	Day 21
UN	0.00 ± 0.00	1.00 ± 0.58	2.67 ± 0.67 ^ab^	4.00 ± 0.37 ^ab^	5.25 ± 0.25 ^abcd^
BC	0.00 ± 0.00	1.17 ± 0.17 ^a^	3.00 ± 0.45 ^ab^	4.40 ± 0.40 ^abc^	6.00 ± 0.00 ^abcd^
EGF	0.00 ± 0.00	0.80 ± 0.37	3.25 ± 0.48 ^ab^	4.67 ± 0.33 ^abc^	5.80 ± 0.20 ^abcd^
TRF	0.00 ± 0.00	2.80 ± 0.20 *^∆a^	3.00 ± 0.58 ^a^	4.60 ± 0.25 ^abc^	5.75 ± 0.25 ^abcd^
EGF + TRF	0.00 ± 0.00	2.29 ± 0.29 *^∆a^	3.40 ± 0.25 ^ab^	6.00 ± 0.26 *^∆abc^	6.67 ± 0.33 *^abc^
SSD	0.00 ± 0.00	1.00 ± 0.00 ^a^	3.67 ± 0.33 ^ab^	4.67 ± 0.21 ^ab^	6.20 ± 0.20 ^abcd^

Note: Values calculated over a 21-day trial and expressed as mean ± S.E.M. for six rats. * *p* < 0.05 compared with those of the UN group on the respective day and ^∆^
*p* < 0.05 compared with those of the SSD group on the respective day. ^a^
*p* < 0.05, ^b^
*p* < 0.05, ^c^
*p* < 0.05 and ^d^
*p* < 0.05 compared to those recorded on days 3, 7, 11 and 14, respectively. Values expressed in a row with distinct superscript differ significantly, at least to a level of 95% confidence (*p* ≤ 0.05).

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
