# Peer review of "Healing Properties of Epidermal Growth Factor and Tocotrienol-Rich Fraction in Deep Partial-Thickness Experimental Burn Wounds"

_antioxidants, 2020, doi:10.3390/antiox9020130_

Round 1
Reviewer 1 Report
This MS is interesting and some assays were performed. However ethical issues must be included.
Other comments:
Line 89 - Thirty six male…
2.2. Animals. In this section it is mentioned twice that animals were housed individually.
Line 196 – Cell Count
Line 231 - dd.H2O?
Line 241 -please indicate Figure 1 and caption in the results section.
Line 264 - Table 2 evidences…
Reviewer 2 Report
The ms of Hui-fang Guo explores the positive effect of the addition of EGF with TRF on thickness burn’s healing parameters.
The ms is not acceptable and some problems should be clarified.
The number of animals for each group is very small, and several results presented in the ms suffer from this problem, with almost unacceptable standars deviations.
Morevoer, the main concern is the total lack of any in vitro characterization of the healing properties as well as any idea about the mechanisms of action. Please clarify the ideas about the combination of these two compounds as well as the molecualr basis of this synergism.
Reviewer 3 Report
Healing Properties of Epidermal Growth Factor and Tocotrienol-Rich Fraction in Deep Partial-Thickness in Experimental Wounds.
The manuscript presents a study in which the effect of EGF+TRF on deep partial-thickness burns was investigated. A number of gross, histological, and biochemical assessments were made. The authors found that treatment with both EGF and TRF had some effect on reepithelization (at and after day Day 14) but a larger effect on dermal repair at day 7 post wounding and an overall greater effect on contraction. To what extent the effects on contraction would be relevant to human burn wound healing is not certain. Based on the results, the authors suggest that TRF may be useful in altering the redox environment of burn wounds and might be useful for accelerating the healing . Burn wounds extending into the reticular region of the dermis are typically debrided to reduce the incidence of infection. This makes determining how the results of this study might translate to humans difficult.
Specific Comments:
Line 3: Delete second <in> Line 89: Please amend the number of animals used in the study Please expand description of thermal source- ie., circular or rectangular surface? Line 138: How long was thermal source applied to skin? The authors noted a roughly doubling of neutrophil numbers from day 11 to 14 and then, these numbers virtually disappear by day 21. Why is this? Wound healing dogma would suggest seeing levels of inflammatory cells peaking at early times and then gradually decline. Example: Lattef et al. 2019 The cutaneous inflammatory response to thermal burn injury in a murine model. Int J Mol Sci 20:538. It might be expected that neutrophil levels and A number of studies have shown that silver inhibits the healing process. The results presented here differ from those findings. This should be noted and discussed. The Discussion contains too much reiteration of the Results and not enough reflection on what these results could mean.
Round 2
Reviewer 2 Report
The authors have not completely clarified the ideas about the combination of these two compounds as well as the molecular basis of this synergism. In particular, they argued that it was difficult to establish a burn model using cellular model. In fact, it is possible to find some results in the scientific literature. However, the authors can try to use a in vitro scratch wound assay in order to demonstrate the synergism.
Round 3
Reviewer 2 Report
The ms can be accepted